# Functionalized Hybrid Iron Oxide–Gold Nanoparticles Targeting Membrane Hsp70 Radiosensitize Triple-Negative Breast Cancer Cells by ROS-Mediated Apoptosis

**DOI:** 10.3390/cancers15041167

**Published:** 2023-02-11

**Authors:** Zhiyuan Wu, Stefan Stangl, Alicia Hernandez-Schnelzer, Fei Wang, Morteza Hasanzadeh Kafshgari, Ali Bashiri Dezfouli, Gabriele Multhoff

**Affiliations:** 1Central Institute for Translational Cancer Research (TranslaTUM), Radiation Immuno Oncology Group, Klinikum Rechts der Isar der Technischen Universität München, 81675 Munich, Germany; 2Department of Nuclear Medicine, Klinikum Rechts der Isar, Technische Universität München, 81675 Munich, Germany; 3Heinz-Nixdorf-Chair of Biomedical Electronics, TranslaTUM, Klinikum Rechts der Isar, Technische Universität München, 81675 Munich, Germany; 4Department of Radiation Oncology, Klinikum Rechts der Isar, Technischen Universität München, 81675 Munich, Germany

**Keywords:** TNBC, Hsp70, TPP-PEG4-FeAuNPs, ROS

## Abstract

**Simple Summary:**

Breast cancer is the most common cancer in women worldwide, and triple-negative breast cancer (TNBC) is the malignancy with the worst prognosis. Although the vast majority of TNBC patients are treated with multimodal therapies, including ionizing irradiation (IR), as a standard of care, radiation-resistant tumor cells and off-target toxicities preclude an advantageous clinical outcome. We studied the radiosensitizing effect of novel Hsp70-specific, hybrid iron oxide–gold (Fe_3_O_4_-Au) nanoparticles (NPs) functionalized with the Hsp70 peptide TPP via a PEG4 linker (TPP-PEG4) to target tumor-specific membrane Hsp70 (mHsp70) on TNBCs. TPP can increase the affinity and uptake of hybrid Fe_3_O_4_-Au nanoparticles (FeAuNPs) into TNBC cells. TPP-PEG4-FeAuNPs, but not control hybrid FeAuNPs, significantly sensitize TNBC cells against radiation by activating a G2/M checkpoint arrest and elevating the production of reactive oxygen species (ROS), which induce DNA double-strand breaks in TNBC.

**Abstract:**

Triple-negative breast cancer (TNBC) a highly aggressive tumor entity with an unfavorable prognosis, is treated by multimodal therapies, including ionizing radiation (IR). Radiation-resistant tumor cells, as well as induced normal tissue toxicity, contribute to the poor clinical outcome of the disease. In this study, we investigated the potential of novel hybrid iron oxide (Fe_3_O_4_)-gold (Au) nanoparticles (FeAuNPs) functionalized with the heat shock protein 70 (Hsp70) tumor-penetrating peptide (TPP) and coupled via a PEG4 linker (TPP-PEG4-FeAuNPs) to improve tumor targeting and uptake of NPs and to break radioresistance in TNBC cell lines 4T1 and MDA-MB-231. Hsp70 is overexpressed in the cytosol and abundantly presented on the cell membrane (mHsp70) of highly aggressive tumor cells, including TNBCs, but not on corresponding normal cells, thus providing a tumor-specific target. The Fe_3_O_4_ core of the NPs can serve as a contrast agent enabling magnetic resonance imaging (MRI) of the tumor, and the nanogold shell radiosensitizes tumor cells by the release of secondary electrons (Auger electrons) upon X-ray irradiation. We demonstrated that the accumulation of TPP-PEG4-FeAuNPs into mHsp70-positive TNBC cells was superior to that of non-conjugated FeAuNPs and FeAuNPs functionalized with a non-specific, scrambled peptide (NGL). After a 24 h co-incubation period of 4T1 and MDA-MB-231 cells with TPP-PEG4-FeAuNPs, but not with control hybrid NPs, ionizing irradiation (IR) causes a cell cycle arrest at G2/M and induces DNA double-strand breaks, thus triggering apoptotic cell death. Since the radiosensitizing effect was completely abolished in the presence of the ROS inhibitor N-acetyl-L-cysteine (NAC), we assume that the TPP-PEG4-FeAuNP-induced apoptosis is mediated via an increased production of ROS.

## 1. Introduction

Breast cancer is the most common cancer worldwide, accounting for approximately 24.5% of all cancers in females, with a mortality-to-incidence ratio of 15.5% [1]. Triple-negative breast cancer (TNBC) a special subtype compromising 10–20% of all breast cancers which does not express estrogen receptors (ER), progesterone receptors (PR), or human epidermal growth factor receptor 2 (HER2) [2]. Compared to other breast cancer types in the same T/N/M stage, the overall survival in patients with TNBC is significantly lower [3]. TNBC is highly aggressive and metastasizes most commonly into the brain and visceral organs; the median survival rate for a metastasized tumor stage is only 13.3 months [4,5]. The lack of targetable receptors makes TNBC insensitive to molecular and endocrine therapies [6]. The vast majority of TNBC patients are treated with multimodal therapies, including ionizing irradiation (IR), as a standard of care, but radiation-resistant tumor cells and off-target toxicities restrict their clinical effectiveness [7]. Therefore, there is a dire medical need for more effective and targeted radiation approaches that can prevent normal tissue toxicity.

The specificity and selectivity of targeted therapies depends on targets which are overexpressed in tumor cells and presented on the cell surface in a tumor-specific manner. Heat shock proteins (HSPs) fulfill these criteria [8,9,10]. Under stress conditions, cells drastically increase the synthesis of HSPs, which, based on their molecular weights, are classified into different families. These include Hsp40, Hsp60, Hsp70, Hsp90 and Hsp110 [11], and the synthesis of other proteins in general is down-regulated. The 72 kDa major stress-inducible Hsp70 (Hsp70, Hsp70–1, HspA1A; #3303) is evolutionarily highly conserved [12] and frequently overexpressed in a wide variety of different tumor entities. In contrast to normal cells, many tumor cell types abundantly present Hsp70 on the plasma membrane; therefore, it serves as a tumor-specific target. A high level of Hsp70 expression in the cytosol and on the plasma membrane positively correlates with the aggressiveness of a tumor [13,14]. A fast uptake of nanoparticles (NPs) targeting mHsp70 is enabled since, compared to other tumor biomarkers such as HER-2, the turn-over rate of membrane-bound Hsp70 into the cytosol is rapid (minute range) [15]. Interestingly, following therapeutic interventions such as radiochemotherapy, the mHsp70 expression is further up-regulated on many different tumor cell types, but not on normal tissues [16]. Due to this tumor-specific and stress-inducible membrane expression of Hsp70 and its fast turn-over rate [15], we considered mHsp70 as an ideal target to enable the uptake of NPs functionalized with an Hsp70-targeting reagent such as cmHsp70.1 monoclonal antibody (mAb). The efficacy of the cmHsp70.1 antibody for the tumor-targeting effect of gold nanoparticles (AuNPs) was recently demonstrated by an increased enrichment and uniform distribution of cmHsp70.1-conjugated NPs in mHsp70-positive tumor cells [12]. In this study, we investigated the capacity of the Hsp70 tumor-penetrating peptide (TPP), instead of the cmHsp70.1 antibody, to target hybrid AuNPs into mHsp70-positive tumor cells.

It is well established that AuNPs can exert radiosensitizing activities on tumor cells via secondary Auger electrons [17]. The low energy of Auger electrons induces a high linear energy transfer (LET) over short distances (nm range) and thereby causes DNA double-strand breaks followed by lethal damage to tumor cells [18]. However, the average dose deposition by AuNPs varies significantly with the distance to the target DNA [19]. Furthermore, the size of the AuNPs greatly influences the radiosensitizing effect. The mean dose linear energy can be increased up to 100% using AuNPs with a size of 2 nm, whereas those with a size of 100 nm increase the mean dose linear energy only up to 40% [19]. 

A new development in nanotechnology is the design of hybrid gold nanosystems such as gadolinium (Gd)-Au, silver (Ag)-Au, palladium (Pd)-Au, cerium oxide (CeO_2_)-Au, and iron oxide (Fe_3_O_4_)-Au NPs. These hybrid NPs were tested successfully in vitro and in preclinical models [20,21,22,23]. To date, Fe_3_O_4_-Au nanoparticles belong to the class of well-studied hybrid NPs that are superior to other NP compositions with respect to their physical properties and biocompatibility [24]. In vivo, the Fe_3_O_4_ core of the NPs serves as a contrast agent, improving magnetic resonance imaging (MRI) of the tumor [25], and the Au shell has the capacity to radiosensitize tumor cells via the release of secondary Auger electrons upon X-ray irradiation [26]. Similarly to other high-Z nanomaterials, AuNPs can enhance the overall tumor absorption of ionizing radiation up to 100-fold. This may alter the type and amount of highly reactive molecules, including ROS [17,27], involved in many important signaling pathways [28,29] which support the induction of DNA double-strand breaks [30]. Previous studies have shown that IR alone induces ROS production, leading to oxidative stress and DNA damage [31,32]. Moreover, ROS production and DNA damage can be further enhanced by combining IR with AuNP therapy [33]. 

In this study, we developed hybrid Fe_3_O_4_-AuNPs (FeAuNPs), functionalized with the Hsp70-targeting peptide TPP and coupled to a 4-mer polyethylene glycol (PEG4) linker (TPP-PEG4-FeAuNPs), to target and kill highly aggressive, mHsp70-positive TNBC cells. TPP is a 14-mer peptide derived from the C-terminal domain of Hsp70 that binds to membrane Hsp70-positive tumor cells. It has a comparable affinity to the full-length cmHsp70.1 antibody because it mimics the binding properties of the oligomerization domain of Hsp70. After binding to mHsp70, the peptide is internalized and subsequently accumulates inside tumor cells [34]. To avoid self-aggregation of the peptide TPP and to improve its stability, a 4-mer PEG chain was implemented into the targeting formulation [35]. Previously, our lab demonstrated that the PET tracer TPP-PEG24-DFO [^89^Zr], which is based on the Hsp70-targeting TPP peptide, binds with a high specificity and selectivity to mHsp70-positive tumor cells [13]. Herein, we tested the binding and uptake capacity and the radiosensitizing effect of hybrid TPP-PEG4-FeAuNPs in the murine and human TNBC cell lines 4T1 and MDA-MB-231.

## 2. Materials and Methods

### 2.1. Hybrid Fe_3_O_4_-Au Nanoparticles (FeAuNPs)

Four (4) nm hybrid nanoparticles were purchased from Nanopartz Inc. Loveland CO, USA. All the hybrid nanoparticles were stored in 5 mM citrate buffer at 4 °C. Fe_3_O_4_ Au-CYS-NHS-PEG4-MAL (Lot#K7310, PEG4-FeAuNPs, 2.5 mg/mL); Fe_3_O_4_ Au-CYS-NHS-PEG4-MAL-PEP (CTKDNNLLGRFELSG) (Lot#K7311, TPP-PEG4-FeAuNPs, 2.3 mg/mL); and Fe_3_O_4_ Au-CYS-NHS-PEG4-MAL-PEP (CNGLTLKNDFSRLEG) (Lot#K7312, NGL-PEG4-FeAuNPs, 2.7 mg/mL) were used.

### 2.2. Cell Culture

TNBC cell lines 4T1 (CRL-2539^TM^, American Type Culture Collection, Manassas, VA, USA, murine) and MDA-MB-231 (HTB-26™, American Type Culture Collection, Manassas, VA, USA, human) were cultured in RPMI 1640 medium (Sigma Aldrich, St. Louis, MO, USA) supplemented with 10% *v/v* heat-inactivated fetal calf serum (FCS) (GIBCO, Eggenstein, Germany) and 1% *v/v* antibiotics (penicillin–streptomycin, GIBCO, Eggenstein, Germany) at 37 °C and 5% CO_2_, at a cell density of 0.5 × 10^6^ cells in 2.5 mL medium. Cells were passaged every second day after seeding. Human peripheral blood lymphocytes (PBL), isolated from EDTA blood of healthy donors using density gradient centrifugation, were grown in supplemented RPMI-1640 in addition to 2 mM L-glutamine (Sigma-Aldrich) and 1 mM sodium pyruvate (Sigma-Aldrich). The cell culture was performed at 37 °C, 95% *v/v* relative humidity, and 5% *v/v* CO_2_. Approval for the taking of blood was obtained by the Institutional Ethical Review Board of the Klinikum rechts der Isar, and written informed consent was obtained from all volunteers.

### 2.3. Reagents

The ROS inhibitors N-actetyl cysteine (NAC) (Cat: ab143032, Abcam, Cambridge, UK) and MG132 (Cat: ab141003, Abcam, Cambridge, UK) were used at concentrations of 5 mM and 300 nM, respectively, and an incubation time of 24 h. DNA double-strand breaks (DSBs) were measured by using the rabbit polyclonal to gamma H2A.X (phospho S139) antibody (Cat: ab11174, Abcam, Cambridge, UK).

### 2.4. Assessment of the Hsp70 Expression on the Cell Membrane

Single cell suspensions of 4T1 and MDA-MB-231 as well as PBL were washed with flow cytometry buffer (PBS/10% *v/v* FCS), followed by an incubation with FITC-conjugated cmHsp70.1 mAb (multimmune GmbH, Munich, Germany) in the dark on ice for 30 min. A FITC-labeled mouse IgG1 (345815, BD Biosciences, Heidelberg, Germany) was used as an isotype-matched control. After two more washing steps, the cells were re-suspended in flow cytometry buffer. Before analysis, using a FACS Calibur instrument (BD Biosciences, Heidelberg, Germany), propidium iodide (PI) was added to each sample. Only PI-negative, viable cells were gated and analyzed. CellQuest Pro 6.0 software was used for data analysis.

### 2.5. Assessment of the Uptake of TPP-Functionalized Nanoparticles into Tumor, but Not Normal, Cells

In order to determine the mHsp70-mediated nanoparticle uptake, mHsp70-positive (MDA-MB-231) and mHsp70-negative PBL were used. NPs were labeled with FITC as described previously [9]. Cells were seeded one day before exposure to the different FITC labeled NP formulations. Flow cytometric analysis was performed 24 h after co-incubation of the cells with either TPP-PEG4-FeAuNPs or NGL-PEG4-FeAuNPs, and after a washing step on the MACSQuant Analyzer 9. Data were compared to untreated control cells.

### 2.6. Visualization of the Internalization of TPP-Functionalized Nanoparticles into Tumor, but Not Normal, Cells

To visualize the NP internalization, tumor (MDA-MB-231) and normal (PBL) cells were seeded overnight and then incubated in fresh complete medium containing FITC-labeled TPP-PEG4-FeAuNPs (2.5 µg/mL). Following a 24 h incubation period at 37 °C, cells were washed twice with cold PBS and then fixed with ice-cold 4% *w*/*v* paraformaldehyde (PFA) in PBS for 15 min at room temperature (RT). After an additional washing step, the filamentous actin (F-actin) and nuclei of the cells were stained with rhodamine-phalloidin (1 μg/mL) and DAPI (2 μg/mL), respectively, for 1 h at RT in the dark. Three-dimensional images of both the tumor (MDA-MB-231) and normal (PBL) cells were acquired using a fluorescence microscope with 63× objective (Leica, THUNDER Imager DMi8, Wetzlar, Germany) equipped with Leica LAS X 3.7.4 software (Leica, Wetzlar, Germany).

### 2.7. Clonogenic Colony Formation Assay (CFA)

Radiosensitivity was determined by using a clonogenic colony formation assay. Briefly, single cell suspensions (500 cells) were seeded into 12-well plates and cultured overnight. Then, cells were incubated with nanoparticles (2.5 µg/mL) for 24 h, followed by X-ray irradiation with 0, 2, 4 and 6 Gy using the Gulmay RS225A irradiation machine (Gulmay Medical Ltd., Camberley, UK; 200 kV, 10 mA, dose rate 1 Gy/min). After 5–7 days, cells were fixed in ice-cold methanol when colonies contained more than 50 cells, then stained with 0.1% crystal violet. All colonies with more than 50 cells were counted. Survival curves were fitted to the linear quadratic model using GraphPad Prism 9 software (GraphPad Software, San Diego, CA, USA), and the survival fraction at each radiation dose was normalized to that of the control without nanoparticles and sham-irradiated with 0 Gy.

### 2.8. Measurement of the Cytotoxicity of Hybrid FeAuNPs in PBL by the CCK-8 Assay

PBL (200,000 cells) were seeded into 96-well u-bottom plates and incubated with different NP formulations for 24 h. Then, cells were centrifuged at 400 rpm for 5 min. The medium was aspirated, and cells were transferred into a 96-well flat-bottom plate and incubated with 100 µL cell culture medium. Next, 10 µL CCK-8 (Cell Counting Kit–8, 96992, Sigma-Aldrich, St. Louis, MI, USA) was added to each well. OD values were measured using a microplate reader at 450 nm at the indicated time points.

### 2.9. Cell Cycle Analysis

TNBCs (4T1, MDA-MB-231) were cultured in 6-well plates overnight, then nanoparticles (2.5 µg/mL) were added for 24 h. Then, the cells were irradiated with 0 and 6 Gy using the Gulmay RS225A irradiation machine (Gulmay Medical Ltd., Cambereley, UK). At 24 h post-irradiation, 5 × 10^5^ cells were collected by trypsinization. Subsequently, cells were fixed in ice-cold methanol (70%) at 4 °C overnight. After a washing step in flow cytometry buffer, cells were incubated in the propidium iodide (PI) staining solution (425 µL 0.1% glucose/PBS, 50 µL RNAse A (0.2 mg/mL)), for 30 min at 37 °C. Before measurement, 25 µL PI (1 mg/mL) was added to each sample. The percentage of cells in a distinct phase of the cell cycle was determined using the FACS Calibur instrument (BD Biosciences, Heidelberg, Germany). CellQuest Pro 6.0 software was used for data analysis.

### 2.10. Annexin V/Propidium Iodide Apoptosis Assay

Apoptosis was analyzed using flow cytometry with the Annexin V-FITC Apoptosis Detection Kit (4830-250-K, R&D Systems, Minneapolis, MN, USA) following the manufacturer’s protocol. Briefly, irradiated or NP-treated/untreated cells were harvested by trypsinization and 2 × 10^5^ cells were washed in PBS. Apoptotic and necrotic cells were analyzed following the addition of 10 μL 10× binding buffer, 10 µL 10× PI, 2 μL fluorescently labeled Annexin V-FITC and 78 µL ddH_2_O in each sample. The samples were mixed gently, incubated at room temperature in the dark for 15 min and then analyzed by flow cytometry on a FACS Calibur instrument. CellQuest Pro 6.0 and/or FlowJo™ (version 10.8.1) software were used for data analysis.

### 2.11. Cellular Reactive Oxygen Species (ROS) Assay

ROS were detected using the 2′,7′-dichlorofuorescin diacetate (DCFDA) Cellular ROS Detection Assay Kit (Cat: ab113851, Abcam, Cambridge, UK). According to the manufacturer’s instructions, 24 h post-irradiation, cells were collected by trypsinization. Then, cells were washed with PBS and incubated with 20 µM DCFDA for 30 min at 37 °C. ROS distribution was determined using the FACS Calibur instrument (BD Biosciences, Heidelberg, Germany). CellQuest Pro 6.0 software was used for data analysis.

### 2.12. Western Blot

Cells were washed with 1× ice-cold PBS, then TBST lysis buffer containing a protease inhibitor was added. Samples were collected in 1.5 mL microtubes on ice, vortexed every 5 min 3 times and centrifuged at 13,000 rpm for 15 min. Protein concentrations were determined using the BCA kit (Cat: 23227, Pierce™ BCA Protein Assay Kit, Thermo Fisher Scientific, MA, USA). The proteins were resolved on a SDS-PAGE and transferred onto NC membrane via semi-dry transfer system (Bio-Rad Laboratories, Inc., CA, USA). Membranes were blocked with 5% skim milk in TBST and incubated with primary antibody overnight at 4 °C. After 3 washing steps, the membranes were exposed to a secondary antibody. Protein bands were developed with chemiluminescence substrate (Cat: 32106, Pierce™ ECL Western Blotting Substrate, Thermo Fisher Scientific, MA, USA). Band intensities were analyzed with image J. The anti-β-actin mAb (Cat: A2228, Sigma-Aldrich, St. Louis, MO, USA) was used as a loading control.

### 2.13. Statistical Analysis

Graphpad Prism software was used for statistical analysis. Data are presented as mean ± SD of *n* = x experiments, with x indicating the number of independent experiments performed. The statistical significance was determined using t-tests with the normally distributed data. Results with *p* <0.05 were considered statistically significant.

## 3. Results

### 3.1. TPP Peptide Increases TNBC’s Affinity to FeAuNPs

The Hsp70 membrane status of viable TNBC cells was determined by flow cytometry using the FITC-conjugated cmHsp70.1 antibody. 4T1 (Figure 1A), and MDA-MB-231 (Figure 1B) cells showed mHsp70-positivity on 63.1 ± 5.68% and 61.94 ± 7.88% of the cells, respectively. Compared to cancer cells, PBL (Figure 1C) showed a mHsp70-positive phenotype only on 3.25 ± 0.49%. Since the Hsp70 peptide TPP has similar binding characteristics to those of the full-length cmHsp70.1 antibody, it was expected that TPP-conjugated NPs would target mHsp70 on tumor cells. The fast turn-over rate of mHsp70 [15] promoted the specific uptake of FeAuNPs by TNBC cells.

Prior to studying the radiosensitizing capacity of hybrid AuNPs, the sublethal concentration of the different peptides’ conjugated and unconjugated NP formulations (TPP-PEG4-FeAuNPs, NGL-PEG4-FeAuNPs, PEG4-FeAuNPs) was determined. When MDA-MB-231 cells were cultured with TPP-PEG4-FeAuNPs at concentrations ranging from 0 up to 20 µg/mL, the internalization of TPP-PEG4-FeAuNPs increased with their concentration (Appendix A) in a clonogenic colony formation assay. No significant differences with respect to the tumor cell viability were observed with either conjugated or unconjugated AuNPs up to a concentration of 2.5 µg/mL (Appendix A). In 4T1 (Appendix A) and PBL (Appendix A) cells, 2.5 µg/mL TPP-PEG4-FeAuNPs also showed no toxic effects. Since all hybrid nanoparticles are stored in 5 mM citrate buffer, the toxicity of the citrate buffer (vehicle) was also determined in a clonogenic colony formation assay. Up to a volume of 20 µL, the citrate buffer showed no toxic effects (Appendix A). Therefore, all further experiments were performed with a NP concentration of 2.5µg/mL in 20 µL citrate buffer. All incubations with NPs were performed at 37 °C 5% CO_2_, because at 4 °C, no incorporation of the NPs was observed.

In the next step, the uptake capacity of the different NP formulations was analyzed comparatively in TNBC cells by microscopy. As illustrated in Figure 2, the highest internalization and most homogeneous distribution of the NPs around the nucleus, compared to unconjugated and control peptide-conjugated NPs, was detected with TPP-PEG4-FeAuNPs in MDA-MB-231 cells after an incubation period of 24 h.

To demonstrate an Hsp70-specific internalization of TPP-PEG4-FeAuNPs into mHsp70-positive MDA-MB-231 tumor cells, but not in mHsp70-negative PBL, both cell types were incubated with FITC-labeled TPP-PEG4-FeAuNPs as described above. As illustrated in Figure 1, MDA-MB-231 cells showed mHsp70 positivity on 63.1 ± 5.68% of the cells, whereas PBL showed nearly no mHsp70 positivity (3.25 ± 0.49%). As a consequence, an uptake of FITC-labeled TPP-PEG4-FeAuNPs into the cytosol occurred predominantly in MDA-MB-231 tumor cells, but not in PBL (Figure 3A,B). A comparison of the quantification of the uptake of the different NP formulations by flow cytometry revealed that 11.9% of the MDA-MB-231 cells were positively stained after incubation with FITC-labeled TPP-PEG4-FeAuNPs, whereas only 1.47% of the cells were stained after incubation with FITC-labeled NGL-PEG4-FeAuNPs for 24 h (Figure 3C). The FITC labeling of both NP formulations was carried out using the same reagents and under identical conditions. As a control, neither the TPP- nor the NGL-functionalized, FITC-labeled NP formulations were taken up into PBL with a low mHsp70 positivity.

### 3.2. TPP-PEG4-FeAuNPs Radiosensitize TNBCs

To study the radiosensitizing effect of the different NP formulations in TNBC cells, 4T1 and MDA-MB-231 cells were incubated with TPP-PEG4-FeAuNPs, NGL-PEG4-FeAuNPs and PEG4-FeAuNPs for 24 h prior to irradiation with 0, 2, 4 and 6 Gy. Clonogenic cell survival was assessed by a colony formation assay. As shown in Figure 4, cells treated with TPP-PEG4-FeAuNPs showed a significantly lower survival rate than sham-treated cells at all radiation doses. In contrast, no significant radiosensitizing effect was induced in TNBC cells pre-treated with NGL-PEG4-FeAuNPs and PEG4-FeAuNPs. These results suggest that the functionalization of AuNPs with the Hsp70 peptide TPP facilitates the internalization and accumulation of FeAuNPs in close proximity to the nuclei of TNBC cells, and thereby sensitizes tumor cells against ionizing irradiation.

### 3.3. TPP-PEG4-FeAuNPs Induce Cell Cycle Arrest at G2/M in TNBCs

It is well accepted that irradiation can cause a cell cycle arrest at the G2/M checkpoint [36]. To study the effects of hybrid AuNPs on cell cycle distribution, 4T1 and MDA-MB-231 cells were cultured with TPP-PEG4-FeAuNPs, NGL-PEG4-FeAuNPs and PEG4-FeAuNPs for 24 h before irradiation. Cell cycle distribution was determined by flow cytometry 24 h after irradiation with 0 and 6 Gy. As shown in Figure 5A,B, an irradiation of 4T1 cells with 6 Gy significantly increased the G2/M ratio from 13.65 ± 2.83% (0 Gy) to 23.95 ± 2.43% (6 Gy). When tumor cells were pre-incubated with TPP-PEG4-FeAuNPs, cells in the G2/M checkpoint increased significantly, up to 31.14 ± 0.44%. Similar results were observed with MDA-MB-231 cells (Figure 5C,D). An irradiation at 6 Gy increased the cell cycle arrest at G2/M from 14.31 ± 4.58% (0 Gy) to 26.36 ± 1.95% (6 Gy). When MDA-MB-231 cells were incubated with TPP-PEG4-FeAuNPs for 24 h before irradiation, the G2/M ratio further increased up to 33.11 ± 1.24%. In contrast, NGL-PEG4-FeAuNPs and PEG4-FeAuNPs did not alter the cell cycle distribution in either the 4T1 or MDA-MB-231 TNBC cells,. This result is in line with the data obtained from the colony formation assay (Figure 4).

### 3.4. TPP-PEG4-FeAuNPs Induce Apoptosis in TNBCs

To address the question of whether apoptosis can be increased in 4T1 and MDA-MB-231 cells by pre-incubation with TPP-PEG4-FeAuNPs before irradiation, Annexin V/PI positivity was determined. As shown in Figure 6A,B, the proportion of apoptotic cells increased up to 1.39 ± 0.15-fold (6 Gy) after irradiation. A pre-incubation with TPP-PEG4-FeAuNPs further increased the proportion of apoptotic cells up to 2.02 ± 0.03-fold in 4T1 cells. In MDA-MB-231 cells (Figure 6C,D), irradiation alone enhanced the proportion of apoptotic cells up to 1.68 ± 0.31-fold (6 Gy), and an additional pre-incubation with TPP-PEG4-FeAuNPs 24 h before irradiation increased it again up to 2.39 ± 0.09-fold. In contrast, pre-incubation with NGL-PEG4-FeAuNPs and PEG4-FeAuNPs did not significantly increase apoptosis in 4T1 or MDA-MB-231 cells.

### 3.5. TPP-PEG4-FeAuNPs Induce Oxidative Stress in TNBCs

To investigate the oxidative stress induced by TPP-PEG4-FeAuNPs, ROS production was measured in MDA-MB-231 cells. NAC is a well-known ROS scavenger which can efficiently inhibit ROS production [37], whereas MG132, a proteasome inhibitor, induces apoptosis by increasing ROS production [38]. Direct binding of MG132 to NAC can reverse the anti-ROS activity of NAC [39]. As illustrated in Figure 7, the ROS production increased in MDA-MB-231 cells 1.32 ± 0.15-fold upon irradiation. When cells were incubated with TPP-PEG4-FeAuNPs before irradiation, the ROS production increased to 2.48 ± 0.18-fold. This effect was reversed when the ROS production was blocked by NAC, and the increased ROS production induced by TPP-PEG4-FeAuNPs dropped to 1.61 ± 0.49-fold. An incubation of the cells with TPP-PEG4-FeAuNPs and the proteasome inhibitor MG132 combined with irradiation increased the ROS production up to 3.32 ± 0.23-fold. This effect was reversed when cells were co-incubated with TPP-PEG4-FeAuNPs, NAC and MG132 followed by irradiation (2.50 ± 0.23-fold). All data were normalized to the values of the 0 Gy sham treatment group.

To further investigate the relationship between ROS production and the radiosensitizing effect of TPP-PEG4-FeAuNPs, identical treatments were performed as described in Figure 7 to determine the cell cycle distribution and apoptosis. As shown in Figure 8A,B, after irradiation, the cell cycle arrest at G2/M increased from 12.05 ± 4.34% (0 Gy) to 25.00 ± 1.08% (6 Gy). When cells were incubated with TPP-PEG4-FeAuNPs for 24 h before irradiation, the G2/M ratio increased to 33.38 ± 2.71%. These data reflect those shown in Figure 5. When ROS production was blocked by NAC, the G2/M ratio dropped to 23.63 ± 2.19 %, and after a combined treatment of TPP-PEG4-FeAuNPs and MG132 followed by irradiation, the G2/M ratio increased up to 42.93 ± 1.50%. This increase in the G2/M ratio could be reversed by a blockade of MG132 by its inhibitor, NAC. The G2/M ratio decreased from 42.93 ± 1.50% to 30.22 ± 0.53%.

With respect to apoptosis (Figure 8C,D), irradiation increased the proportion of apoptotic cells to 2.11 ± 0.64-fold. When cells were additionally pre-incubated with TPP-PEG4-FeAuNPs for 24 h, the proportion of apoptotic cells increased from 2.11 ± 0.64-fold to 3.30 ± 0.28-fold. When ROS production induced by TPP-PEG4-FeAuNPs was blocked by NAC, apoptosis decreased to 2.07 ± 0.48-fold. This effect was further increased when TPP-PEG4-FeAuNPs and irradiation were combined with MG132; the proportion of apoptotic cells increased to 4.81 ± 0.59-fold. However, when the MG132 effect was blocked by NAC, cells produced fewer ROS, and the proportion of apoptotic cells decreased to 2.31 ± 0.45-fold. A comparison of the results derived from the cell cycle, apoptosis and ROS production analyses revealed that the cell cycle arrest at the G2/M checkpoint, in apoptosis and in ROS production followed the same trend. These data indicate that the TPP-PEG4-FeAuNPs-induced radiosensitization effect might be dependent on G2/M checkpoint arrest and increased ROS production.

### 3.6. TPP-PEG4-FeAuNPs Induce DNA Double-Strand Breaks in TNBCs

It is well known that ionizing irradiation causes DNA double-strand breaks [40]. To investigate the effects of TPP-PEG4-FeAuNPs on DNA double-strand breaks, γ-H2AX was determined in MDA-MB-231 cells exposed to the different treatment schedules by Western blot. As shown in Figure 9, irradiation induced a 2.08 ± 0.26-fold increase in MDA-MB-231 cells. When cells were pre-incubated with TPP-PEG4-FeAuNPs for 24 h before irradiation, the γ-H2AX levels increased to 3.11 ± 0.37-fold. After a NAC blockade of the ROS production, the γ-H2AX levels decreased to 1.52 ± 0.79-fold. The highest increase in γ-H2AX levels (6.24 ± 0.14-fold) was observed when TPP-PEG4-FeAuNPs were combined with MG132 and irradiation. This effect could be inhibited when the activity of the protease inhibitor MG132 was blocked by NAC, the γ-H2AX levels dropped from 6.24 ± 0.14-fold to 2.55 ± 0.98-fold. In summary, the γ-H2AX levels, which are indicative of a DNA double-strand break repair, correlate with the ROS production.

## 4. Discussion

Radiotherapy is widely used in the multimodal treatment concept of TNBC. However, side effects caused by radiation-induced off-target toxicity reduce its clinical efficacy [7]. To reduce radiation-induced normal tissue toxicity, especially in the lung and in the heart [41,42], ionizing irradiation has been combined with high Z metal-based nanoparticle therapies [43]. Iron, silver, iodine and gold are used especially frequently for the production of high Z metal nanoparticles [44,45,46,47]. AuNPs enhanced radiation therapy for squamous cell carcinoma [48] and mammary carcinoma [49] in mice and showed a high clearance capacity via the kidneys [49]. In an effort to further improve nanoparticle-based approaches, hybrid NPs were developed that combined the beneficial biophysical properties and functions of the different materials. Fe_3_O_4_-Au hybrid nanoparticles are presently the best-studied, since the Fe_3_O_4_ core plays a key role as an MRI contrast-enhancing agent and the gold shell can exert radiosensitizing activities via secondary Auger electrons [50]. By coupling tumor-targeting agents such as antibodies or peptides to the surface of the nanoparticles, the targeting capability can be improved. Human mesenchymal stem cells [51], peptide [52] and antibody [53] functionalized nanoparticles showed a high affinity to tumors in mice.

In our study, we utilized the 14-mer Hsp70 peptide TPP for tumor-specific targeting of highly aggressive, mHsp70-positive TNBC cells. TPP conjugated to hybrid FeAuNPs enabled an improved uptake and a more homogenous distribution of the NPs around the nuclei of the tumor cells [34]. Since normal cells lack a membrane Hsp70 expression, they most likely will not internalize TPP-functionalized NPs. Because the sequence mimics properties of the Hsp70 oligomerization domain, TPP shows a strong self-aggregation capacity [34]. To improve the stability, solubility and biodistribution kinetics of FeAuNPs, a 4-mer PEG chain was introduced to the binding-active TPP moiety. The PEG chain can avoid self-aggregation of the peptide TPP [35,54]. The diameter of the hybrid TPP-PEG4-FeAuNPs after functionalization with TPP was only 4 nm. A previous study indicated that small-sized NPs more effectively penetrate the nucleus [55]. As internal controls, we developed FeAu hybrid nanoparticles functionalized with a nonspecific 14-mer scrambled peptide (NGL-PEG4-FeAuNPs), as well as nanoparticles without a peptide but with the PEG4 linker (PEG4-FeAuNPs).

Before treatment of TNBCs with hybrid nanoparticles, we checked the Hsp70 membrane status of the TNBCs 4T1 and MDA-MB-231. As expected, both highly aggressive tumor cell types showed a strong mHsp70 expression, which is a prerequisite for the therapy with TPP-functionalized NPs that target mHsp70 on tumor cells. As a control, normal cells such as PBL were used, as they have a very low mHsp70 expression. In previous studies, we demonstrated a rapid internalization of Hsp70-targeting probes, such as cmHsp70.1 antibody [56] and TPP [34], into the cytoplasm of mHsp70-positive tumor cells. In contrast to control AuNPs, the cmHsp70.1 antibody functionalized AuNPs significantly enriched inside tumor cells both in vitro and in vivo [12]. The tumor affinity of the different NP formulations was tested in MDA-MB-231 cells and PBL. Tumor cells (MDA-MB-231) with high mHsp70 positivity show significant enrichment of TPP-PEG4-FeAuNPs in the cytosol, whereas PBL with very low mHsp70 positivity showed no uptake or enrichment of this NP formulation. In MDA-MB-231 tumor cells, TPP-PEG4-FeAuNPs are internalized and more homogeneously distributed around the nucleus compared to NGL-PEG4-FeAuNPs. AuNPs in close proximity to the nuclei of tumor cells significantly enhance their radiosensitivity [57]. Because of the short range (2–500 nm) of secondary Auger electrons [58], the accumulation of FeAuNPs in close proximity to the nucleus is critical to initiating DNA damage. Thus, the radiosensitizing effect of the different NP formulations, including the TPP-PEG4-FeAuNPs, was tested in TNBC cells 4T1 and MDA-MB-231 in a clonogenic cell survival assay. After irradiation with different doses, clonogenic cell survival was inhibited in a dose-dependent manner. Pre-treatment with TPP-PEG4-FeAuNPs before irradiation significantly increased the tumor cell death rate, whereas the control NP formulations (FeAuNPs with a non-specific scrambled peptide or without a peptide) did not show any significant effect. These results suggest that the affinity and uptake of FeAuNPs can be improved by functionalization with the mHsp70-targeting peptide TPP.

The G2/M checkpoint is crucial for eukaryotic cells to prevent cell mitosis in cells with damaged DNA [59]. Radiation induces cell cycle arrest at the radiosensitive G2/M phase in TNBC cells 4T1 and MDA-MB-231 [60]. In our study, the checkpoint arrest was studied in TNBC cells after irradiation with 6 Gy and in combination with hybrid AuNPs. Irradiation alone induced a G2/M arrest in both TNBC cells. After treatment with TPP-PEG4-FeAuNPs before irradiation, the G2/M arrest was more pronounced than after irradiation alone. The lower affinity and uptake of non-functionalized control AuNPs caused weaker effects with respect to the G2/M arrest. The G2/M arrest increased the efficiency of internalization of NPs [61] and nuclear uptake [62]. The G2/M cell cycle arrest is related to cell apoptosis [63]. In G2/M phase, the anti-apoptotic Bcl-2 protein is phosphorylated [64]. Our results confirmed this finding in 4T1 and MDA-MB-231 cells. The highest apoptosis rate was achieved in TNBCs after treatment with TPP-PEG4-FeAuNPs followed by irradiation.

Endogenous ROS are continuously generated along the mitochondria electron transport chain [65]. They involved in many important signaling pathways, including cell cycle arrest and apoptosis [66,67]. Irradiation can induce ROS production in multiple cancer cell types [68,69]. The irradiation induced ROS production by sensing Ca^2+^ channels, which are linked to salivary gland dysfunction in mice [70]. In our study, we used both NAC, an important antioxidant [71], and MG132, a proteasome inhibitor, to support the degradation of contents involved in the antioxidant defense system [72] and to explore radiation-induced oxidative stress in TNBC cells. We chose DCFDA/H_2_DCFDA to determine ROS in our cell types. DCFDA/H_2_DCFDA does not detect specific types of ROS, but is a general indicator of ROS such as superoxide, hydrogen peroxide, and others [73]. In our experiments, irradiation increases ROS production in MDA-MB-231 cells. When cells were pre-incubated with TPP-PEG4-FeAuNPs as a radiosensitizer, the ROS production could be significantly enhanced. In an effort to demonstrate the association between ROS production and the radiosensitizing effect, we applied the ROS inhibitor NAC. We showed that NAC blocks not only the ROS production induced by irradiation and TPP-PEG4-FeAuNPs in MDA-MB-231 cells, but also apoptotic cell death. In our study, we were able to demonstrate that apoptotic cell death induced by TPP-PEG4-FeAuNPs and followed by irradiation depends on G2/M cell cycle arrest and increased ROS production.

High ROS levels are closely related to irradiation-induced DNA double-strand breaks [32]. In the study by X. Qin, biogenetic gold nanoparticles (Au@MC38) resulted in ROS generation and DNA damage after irradiation [33]. We used Western blot analysis to demonstrate the link between elevated ROS levels and increased DNA double-strand break frequency as determined by a higher γ-H2AX expression, which is indicative for DNA double-strand repair. Our results showed an increased γ-H2AX expression in MDA-MB-231 cells 1 h after irradiation and a pre-treatment with functionalized hybrid FeAuNPs, which was closely related to the elevated ROS production. The DNA damage effect was further enhanced by the addition of the protease inhibitor MG132, and was completely reversed by the ROS inhibitor NAC. In a study of H. Wang, NAC was shown to block ROS in MDA-MB-231 cells, but did not decrease the anticancer efficacy of X-ray irradiation [74]. Auger electrons released by AuNPs upon X-ray irradiation work only at a short distance to the target (2–500 nm) [58]. Therefore, the TPP-PEG4-FeAuNPs that were found to be accumulated in close proximity to the nucleus did, in fact, radiosensitize tumor cells. However, it remains to be elucidated how the TPP peptide enables the transport of NPs to the nucleus inside tumor cells.

## 5. Conclusions

We studied the radiosensitizing effect of novel TPP-PEG4-based FeAu hybrid nanoparticles functionalized with the Hsp70 peptide TPP to target mHsp70-positive TNBCs. TPP increased the affinity and uptake of FeAuNPs into TNBC cells. TPP-PEG4-FeAuNPs, but not control hybrid FeAuNPs, significantly sensitized TNBC cells against radiation by a G2/M checkpoint arrest, as well as increased ROS production which, in turn, caused ROS-dependent DNA damage. Further in vivo studies are necessary to explore the effects of the TPP-functionalized hybrid FeAuNPs in clinically relevant mouse tumor models in combination with ionizing irradiation.

## Figures and Tables

**Figure 1 cancers-15-01167-f001:**
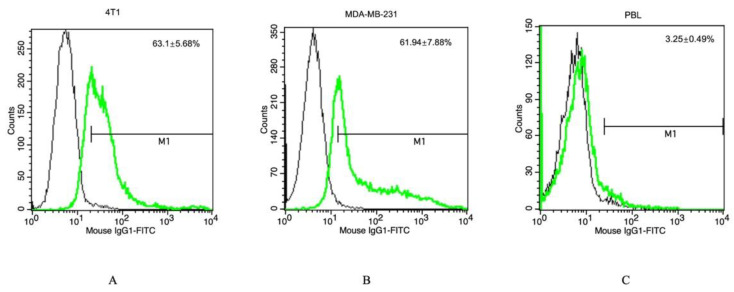
Representative histograms of a flow cytometric analysis of 4T1 (**A**), MDA-MB-231 (**B**) and PBL (**C**) using FITC-labeled cmHsp70.1 antibody (green line) and an isotype-matched control antibody (black line). Values in the upper right corner of each graph represent mean values ± SD of 3 independent experiments.

**Figure 2 cancers-15-01167-f002:**
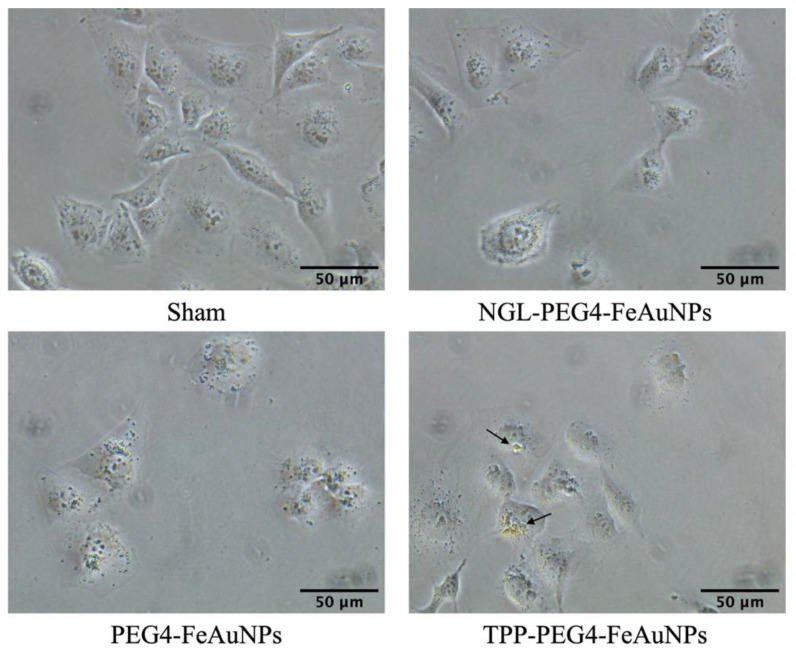
Representative images of MDA-MB-231 cells cultured with TPP-PEG4-FeAuNPs, PEG4-FeAuNPs and NGL-PEG4-FeAuNPs (2.5 µg/mL for 24 h), or after sham treatment. Black arrows show the NP accumulation in MDA-MB-231 cells. Images were recorded with color camera images of transmitted visible light using a 40× objective.

**Figure 3 cancers-15-01167-f003:**
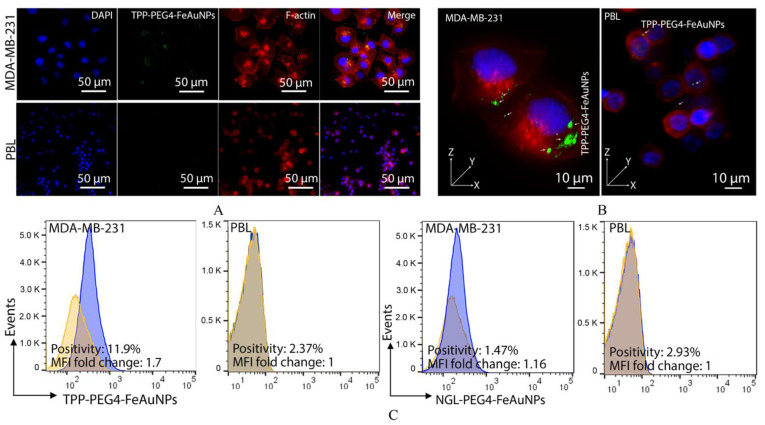
(**A**,**B**) Visualization of the specific uptake of TPP-PEG4-FeAuNPs into mHsp70-positive MDA-MB-231 tumor cells, but not into mHsp70-negative normal cells (PBL). DAPI was shown as blue, green for FITC and red for F-actin. (**B**) 3D images of the tumor (MDA-MB-231) and normal (PBL) cells were acquired by fluorescence microscopy. (**C**) Quantification of the uptake of TPP-PEG4-FeAuNPs and NGL-PEG4-FeAuNPs into mHsp70-positive tumor cells (MDA-MB-231) and normal cells (PBL) was determined by flow cytometry. On Y axis, k is equivalent to 1000. Experiments were repeated twice and representative data are presented.

**Figure 4 cancers-15-01167-f004:**
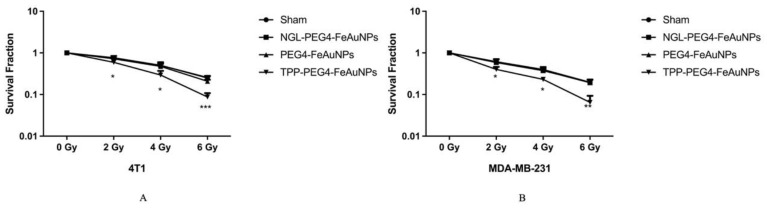
Clonogenic cell survival of 4T1 (**A**) and MDA-MB-231 (**B**) TNBC cells after treatment with NGL-PEG4-FeAuNPs, PEG4-FeAuNPs and TPP-PEG4-FeAuNPs (each 2.5 µg/mL for 24 h) or a sham treatment followed by an irradiation with 0, 2, 4 and 6 Gy. Significant differences were observed between the sham and TPP-PEG4-FeAuNP-treated cells. Significance: * *p* ≤ 0.05, ** *p* ≤ 0.01, *** *p* ≤ 0.001; results represent the mean values ± SD of 3 independent experiments.

**Figure 5 cancers-15-01167-f005:**
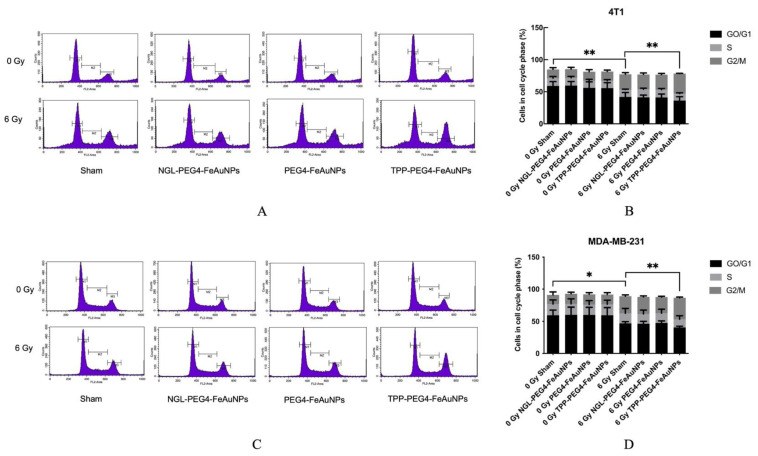
Cell cycle analysis of 4T1 (**A**,**B**) and MDA-MB-231 (**C**,**D**) TNBC cells after treatment with NGL-PEG4-FeAuNPs, PEG4-FeAuNPs or TPP-PEG4-FeAuNPs for 24 h followed by an irradiation with 6 Gy. TPP-PEG4-FeAuNPs induce significant cell cycle arrest at the G2/M checkpoint. Cell cycle distribution was measured in permeabilized cells by flow cytometry. Significance: * *p* ≤ 0.05, ** *p* ≤ 0.01; results represent mean values ± SD of 3 independent experiments.

**Figure 6 cancers-15-01167-f006:**
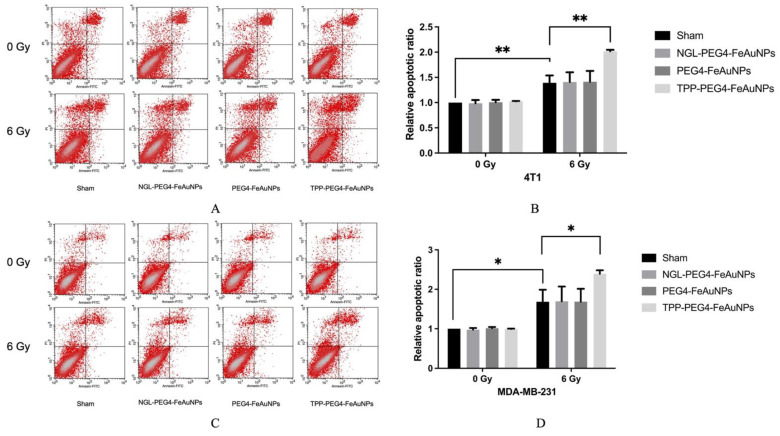
Apoptosis assay of 4T1 (**A**,**B**) and MDA-MB-231 (**C**,**D**) TNBC cells after treatment with NGL-PEG4-FeAuNPs, PEG4-FeAuNPs and TPP-PEG4-FeAuNPs, or sham treatment for 24 h followed by an irradiation with 6 Gy. Apoptosis was determined by flow cytometry using Annexin V-FITC/PI staining. Significance: * *p* ≤ 0.05, ** *p* ≤ 0.01; results show the mean values ± SD of 3 independent experiments.

**Figure 7 cancers-15-01167-f007:**
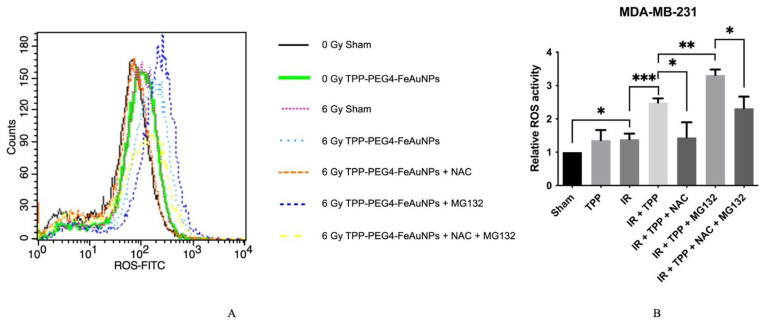
Analysis of the ROS production in MDA-MB-231 TNBC cells (**A**,**B**) after treatment with TPP-PEG4-FeAuNPs for 24 h, either alone or in combination with the ROS inhibitors NAC (5 mM) and MG132 (300 nM) and irradiation with 0 Gy and 6 Gy. ROS production was measured by DCFDA staining over 24 h by flow cytometry. Abbreviations: IR, irradiation; TPP, TPP-PEG4-FeAuNPs; NAC, N-Acetyl-L-cysteine; MG132, Z-Leu-Leu-Leu-al. Significance: * *p* ≤ 0.05, ** *p* ≤ 0.01, *** *p* ≤ 0.001. Results represent the mean values ± SD of 3 independent experiments.

**Figure 8 cancers-15-01167-f008:**
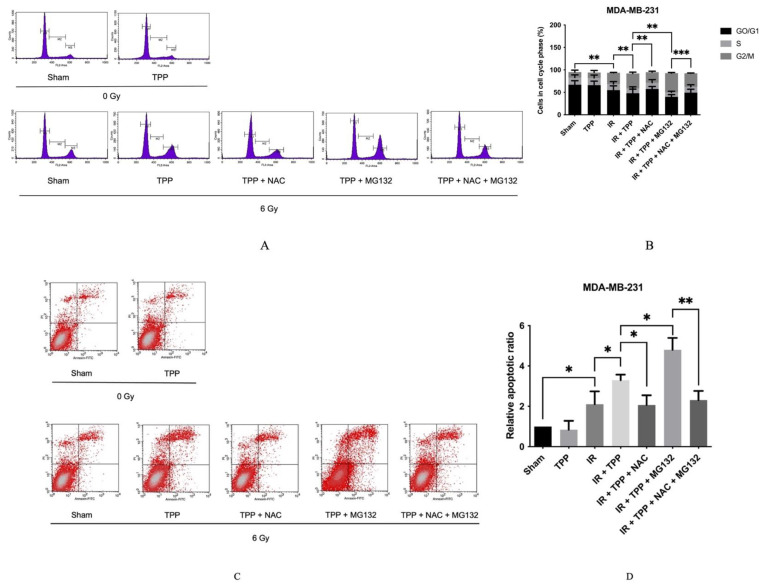
Analysis of the cell cycle arrest and apoptosis in MDA-MB-231 TNBC cells after treatment with TPP-PEG4-FeAuNPs, either alone or in combination with ROS inhibitors NAC (5 mM) and MG132 (300 nM), followed by irradiation with 0 Gy and 6 Gy. Cell cycle (**A**,**B**) and Annexin V-FITC/PI staining (**C**,**D**) were measured by flow cytometry. Abbreviations: IR, irradiation; TPP, TPP-PEG4-FeAuNPs; NAC, N-acetyl-L-cysteine; MG132, Z-Leu-Leu-Leu-al. Significance: * *p* ≤ 0.05, ** *p* ≤ 0.01, *** *p* ≤ 0.001. Results show the mean values ± SD of 3 independent experiments.

**Figure 9 cancers-15-01167-f009:**
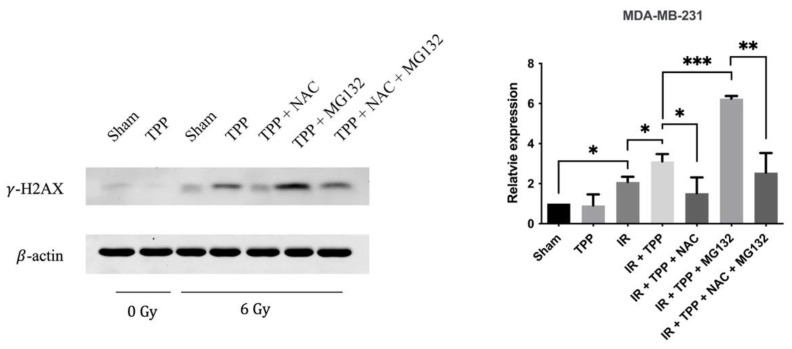
DNA DSBs induced by TPP-PEG4-FeAuNPs are dependent on ROS production in MDA-MB-231 cells. MDA-MB-231 cells were treated as described in Figure 7. Cells were harvested 1 h after irradiation for Western blotting. γ-H2AX was used to determine DNA DSBs. β-actin was used as a loading control. The uncropped Western Blot images can be found in Appendix A. Abbreviations: IR, irradiation; TPP, TPP-PEG4-FeAuNPs; NAC, N-Acetyl-L-cysteine; MG132, Z-Leu-Leu-Leu-al. Significance: * *p* ≤ 0.05, ** *p* ≤ 0.01, *** *p* ≤ 0.001; results represent the mean values ± SD of 3 independent experiments. The uncropped Western Blot images can be found in Appendix A.

## Data Availability

The data presented in this study are available form authors.

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
