# Peer review of "Functionalized Hybrid Iron Oxide–Gold Nanoparticles Targeting Membrane Hsp70 Radiosensitize Triple-Negative Breast Cancer Cells by ROS-Mediated Apoptosis"

_cancers, 2023, doi:10.3390/cancers15041167_

Round 1
Reviewer 1 Report (Previous Reviewer 1)
The authors have carried out the suggested changes and the manuscript is now suitable for publication in Cancers
Author Response
Dear Reviewer,
The authors want to thank you for taking time to review our paper. We appreciate your constructive comments.
King regards,
Zhiyuan Wu
Reviewer 2 Report (New Reviewer)
Manuscript ID cancers-2201221 describes the potential of TPP-PEG4-based FeAu hybrid nanoparticles functionalised with the Hsp70 TPP peptide for the treatment of radiosensitising triple-negative breast cancer. The authors demonstrated that TPP (mHsp70-targeting peptide) can improve the affinity and uptake of hybrid iron oxide and gold nanoparticles into cells. Investigation of the mechanism of action showed that the hybrid nanoparticles induce cell cycle arrest and increased ROS production and induce DNA double-strand breaks. The research is very interesting and I propose to publish it after a minor correction.
Please improve formatting - sentences appear coloured red or blue in the text.
The authors should provide precise information on the method used to calculate statistical significance in the experimental section. In addition, the authors should add in the experimental section the number of times the experiments were repeated.
Poor resolution of figure 3 - authors should present better photos. Most photos do not include a scale bar. Please add a scale bar everywhere. The histograms in Figure 3 are also not clear.
Graphs in Figures 5 and 6 - the authors should enlarge the size to make the axes more visible.
Did the authors conduct studies targeting antioxidant defence or induction of repair systems? In my opinion, the authors could have been tempted to investigate the mechanism of action more extensively, aiming to understand the pathway of apoptosis induction or oxidative stress induction.
Author Response
Dear Reviewer,
Thank you for taking time to review our paper, following are the answers to your question.
Point 1: Please improve formatting - sentences appear coloured red or blue in the text.
Response 1: Thank you for suggestion, we’ve changed the font to black.
Point 2: The authors should provide precise information on the method used to calculate statistical significance in the experimental section. In addition, the authors should add in the experimental section the number of times the experiments were repeated.
Response 2: Our data are presented as mean± SD of n=x experiments, with x indicating the number of independent experiments performed. The statistical significance was determined using t-test with the data which are normally distributed.
Point 3: Poor resolution of figure 3 - authors should present better photos. Most photos do not include a scale bar. Please add a scale bar everywhere. The histograms in Figure 3 are also not clear.
Response 3: We changed figure 3 with a better resolution and add scale bar to all the photos.
Point 4: Graphs in Figures 5 and 6 - the authors should enlarge the size to make the axes more visible.
Response 4: We enlarged graphs in Figures 5, 6 and 8 to make the axes more visible.
Point 5: Did the authors conduct studies targeting antioxidant defence or induction of repair systems? In my opinion, the authors could have been tempted to investigate the mechanism of action more extensively, aiming to understand the pathway of apoptosis induction or oxidative stress induction.
Response 5: Thank you for your question about studies targeting antioxidant defense or induction of repair systems. In our study, we used both NAC an important antioxidant (Li and Zhao 2019) and MG132 a proteasome inhibitor that supports the degradation of contents involved in the antioxidant defense system(Han, Moon et al. 2009) to explore radiation-induced oxidative stress in TNBC cells. The results showed that NAC not only blocks the ROS production induced by irradiation and TPP-PEG4-FeAuNPs in MDA-MB-231 cells, but also the apoptotic cell death, cell cycle and DNA damage effect. After direct binding of MG132 to NAC (Halasi, Wang et al. 2013) the activity of MG132 is blocked, the ROS production is reduced and apoptosis is thereby avoided.
Reference
Halasi, M., M. Wang, T. S. Chavan, V. Gaponenko, N. Hay and A. L. Gartel (2013). "ROS inhibitor N-acetyl-L-cysteine antagonizes the activity of proteasome inhibitors." Biochem J 454(2): 201-208.
Han, Y. H., H. J. Moon, B. R. You and W. H. Park (2009). "The effect of MG132, a proteasome inhibitor on HeLa cells in relation to cell growth, reactive oxygen species and GSH." Oncol Rep 22(1): 215-221.
Li, Q. and Z. Zhao (2019). "Influence of N-acetyl-L-cysteine against bisphenol a on the maturation of mouse oocytes and embryo development: in vitro study." BMC Pharmacol Toxicol 20(1): 43.
The authors want to thank reviewer for constructive comments.
This manuscript is a resubmission of an earlier submission. The following is a list of the peer review reports and author responses from that submission.
Round 1
Reviewer 1 Report
This manuscript is very interesting and is focused on a hot current topic as the use of nanotechnology and nanomaterials in cancer treatment, especially TNBC.
The manuscript is well written and clear enough to be understood and the results are sound and deserve publication in Cancers, however, there are some points of criticism of this reviewer that need to be addressed:
1) In the images given in Figure 2 one can infere the internalization of the NPs, however, it could improve the quality of the manuscript to do additional studies to have a 3D images model to confirm that is not overlapping but inside the cells.
2) The results are sound but is not clear how is the behaviour of the studied systems when using normal cells, epithelial cells could be OK, how is the selectivity, how is the toxicity in normal conditions? Maybe I am wrong but I could not see that in the manuscript.
3) The study does not differenciate the type of ROS that are promoted by the studied systems, it could be interesting to see whether these systems are promoting the formation of OH· or, on the other hand superoxide radicals or peroxides are there, there are several studies in solution (simulating a physiological medium) that could be carried out to study this and this would improve significantly the quality of the manuscript.
If these corrections/additions are carried out, the manuscript will become suitable for publication.
Author Response
Dear Reviewer,
Thank you for your suggestions. Following is our answers to your comments:
1) In the images given in Figure 2 one can infere the internalization of the NPs, however, it could improve the quality of the manuscript to do additional studies to have a 3D images model to confirm that is not overlapping but inside the cells.
Answer: Thank you for your suggestion. Before taking images of the cells, we washed the cells with PBS to deplete extracellular NPs. A 3D model would be perfect for our research to provide more detailed information about the internalization of NPs. But it will take several weeks to perform these experiments. In the future we will perform this experiment.
2) The results are sound but is not clear how is the behaviour of the studied systems when using normal cells, epithelial cells could be OK, how is the selectivity, how is the toxicity in normal conditions? Maybe I am wrong but I could not see that in the manuscript.
Answer: Our research focusses on the radiosensitizing effects of NPs on TNBC. Two TNBC cell lines 4T1 and MDA-MB-231 with high expression of mHsp70 were used. The 14-mer Hsp70 peptide TPP has a tumor-specific targeting of highly aggressive, mHsp70-positive TNBC cells. Since normal cells lack a membrane Hsp70 expression they most likely will not bind and internalize TPP functionalized NPs. Previous in vivo imaging data of our group clearly demonstrate that labeled TPP peptide does not bind to any normal tissues including epithelial cells, but exclusively bind to tumor cells expressing Hsp70 on the cell membrane (Stangl et al Can Res 2018). Furthermore, we could show that the binding and internalization of labeled TPP is depending on the membrane Hsp70 density. Since normal cells do not present Hsp70 on the cell surface binding and uptake of TPP-functionalized NPs is not very likely.
3) The study does not differenciate the type of ROS that are promoted by the studied systems, it could be interesting to see whether these systems are promoting the formation of OH· or, on the other hand superoxide radicals or peroxides are there, there are several studies in solution (simulating a physiological medium) that could be carried out to study this and this would improve significantly the quality of the manuscript.
Answer: Irradiation can induce several types of ROS productions (Riley PA. Int J Radiat Biol. 1994). The kit DCFDA/H2DCFDA which we used in our study is detecting different types of ROS and cannot specify different subclasses. NAC is also a general inhibitor of the ROS, but not against a specific type. So, we cannot differentiate a specific type of ROS.
The autrhors want to thank the reviewer for helpful suggestions.
King regards,
Wu

Reviewer 2 Report
Observations and comments to the manuscript:
“Functionalized Hybrid Gold-Iron Oxide Nanoparticles Targeting Membrane Hsp70 Radiosensitize Triple-negative Breast Cancer Cells by ROS-mediated Apoptosis”
The manuscript is interesting, and requires careful review on the following points:
1) Clearly demonstrate that NTT-functionalized nanoparticles enter cells, that this entry is specific, and that there is accumulation of the nanoparticles within cells.
2) Clonogenic survival results show inconclusive results, because NGL and PEG-4 nanoparticles also lower their survival.
3) In Figure 2, the authors mention the accumulation of TTP-functionalized nanoparticles in MDA-MB231 cells, however it is not clear and the evidence is not solid, it might be better to use electron microscopy to visualize them, confocal microscopy or flow cytometry assays with the fluorophore-labeled nanoparticles.
4) In Figure S2, the clonogenic cell survival results are shown, and the results are similar for the NGL and TPP nanoparticles. A percentage of MDA-MB3-231 and 4T1 cells do not carry HSP70 in the membrane (31%), could it be that these cells are the survivors because they do not take up the TTP-nanoparticles?
5) Figures 4B and 4D show graphs of percentages of cell cycle phases and the y-axis mentions percentage of nucleotides? A careful review of the figures is suggested.
6) In Figure 5, the treatments with 0 and 6 Gy of radioactivity on MDA-MB-231 cells show increases with NGL and PEG4 nanoparticles, which theoretically should not have an effect on the apoptosis process, and the difference with respect to nanoparticles with TPP does not seem to be significant.
The results are in the same direction as the experiments in the previous figures.
6) The results in Figure 7C where the percentages of apoptotic cells are compared seem contradictory, the treatment with TPP vs TPP+NAC at 6 Grys, does not seem to correlate with the graphs in 7D, because the TPP bar is lower but in the flow cytometry analysis the population is higher in TPP.
Author Response
Dear Reviewer,
Thank you for your suggestions. Following is our answers to your comments:
1)Clearly demonstrate that NTT-functionalized nanoparticles enter cells, that this entry is specific, and that there is accumulation of the nanoparticles within cells.
Answer: Thank you for your this suggestion. Electron microscopy will be perfect for our research to provide more detailed information on the internalization of the NPs. But it will takes several weeks to perform these experiments. In a future project we plan to do EM.
2) Clonogenic survival results show inconclusive results, because NGL and PEG-4 nanoparticles also lower their survival.
Answer: Compared with the irradiation group, cells incubated with NGL-PEG4-FeAuNPs and PEG4-FeAuNPs for 24 h prior to irradiation showed no statistical significant differences in cell viability.
3) In Figure 2, the authors mention the accumulation of TTP-functionalized nanoparticles in MDA-MB231 cells, however it is not clear and the evidence is not solid, it might be better to use electron microscopy to visualize them, confocal microscopy or flow cytometry assays with the fluorophore-labeled nanoparticles.
Answer: Thank you for this suggestion. Before taking images of the cells, we washed the cells with PBS to deplete extracellular NPs. Electron microscopy will be perfect for our research to prove the internalization of the NPs. But it will take several weeks to perform these experiments. Unfortunately, our nanoparticles were not fluorophor-labeled, so it was not possible to do confocal microscopy or flow cytometry analysis.
4) In Figure S2, the clonogenic cell survival results are shown, and the results are similar for the NGL and TPP nanoparticles. A percentage of MDA-MB3-231 and 4T1 cells do not carry HSP70 in the membrane (31%), could it be that these cells are the survivors because they do not take up the TTP-nanoparticles?
Answer: Hsp70 played an important role in cancer cell survival, tumorigenicity and anti-apoptotic activities. Cancer cells with Hsp70 knockout or knockdown are more sensitive to irradiation (Murakami N, Radiat Oncol. 2015). TNBC is less sensitive to radiation (Kyndi M, J Clin Oncol. 2008). In our study, the hybrid Fe3O4-AuNPs functionalized with the Hsp70-peptide TPP (TPP-PEG4-FeAuNPs) target TNBC cells and significantly sensitized TNBC cells against radiation. We believe cells survived from radiation due to their resistance to radiation but not the absence of Hsp70.
5) Figures 4B and 4D show graphs of percentages of cell cycle phases and the y-axis mentions percentage of nucleotides? A careful review of the figures is suggested.
Answer: Thank you for your suggestion, the y-axis represents the cells in a distinct cycle phase (%). We changed the description of the Y axis in the Ms.
6) In Figure 5, the treatments with 0 and 6 Gy of radioactivity on MDA-MB-231 cells show increases with NGL and PEG4 nanoparticles, which theoretically should not have an effect on the apoptosis process, and the difference with respect to nanoparticles with TPP does not seem to be significant.
The results are in the same direction as the experiments in the previous figures.
Answer: Irradiation can induce a lower survival of 4T1 and MDA-MB-231 cells which was shown in Figure 3. In Figure 5, we show that irradiation induces apoptosis which is consistent with the results of the colony formation assay in Figure 3. A pre-incubation with NGL-PEG4-FeAuNPs and PEG4-FeAuNPs did not increase apoptosis compared to irradiation alone. But a pre-incubation with TPP-PEG4-FeAuNPs increased the proportion of apoptotic cells significantly compared to irradiation alone.
7) The results in Figure 7C where the percentages of apoptotic cells are compared seem contradictory, the treatment with TPP vs TPP+NAC at 6 Grys, does not seem to correlate with the graphs in 7D, because the TPP bar is lower but in the flow cytometry analysis the population is higher in TPP.
Answer: In both Figure 7C and 7D, the IR+TPP group showed a higher proportion of apoptosis (3.30±0.28-fold), and in the IR+TPP+NAC group apoptosis was found to be decreased (2.07±0.48-fold).
The authors want to thank both reviewers for constructive comments.
Best wishes,
Wu

Round 2
Reviewer 1 Report
The authors did not do any of the suggested experiments. I consider necessary to do a 3D model of internalization and a study of the toxicity in normal cells (not only basing the comment on literature studies).
I still ask for these revisions to suggest acceptance.
Author Response
Dear Reviewer,
With your suggestions, we did 3D images and use PBL cells for further experiments. Here's the answer to your questions with our new results.
1. In the images given in Figure 2 one can infere the internalization of the NPs, however, it could improve the quality of the manuscript to do additional studies to have a 3D images model to confirm that is not overlapping but inside the cells.
Answer: Thank you for your suggestion. In the further experiments, we labeled NPs with FITC. Visualization of the specific uptake of TPP-PEG4-FeAuNPs into mHsp70-positive MDA-MB-231 cells was shown in Figure 3. As shown in Fig3A, FITC-labeled TPP-PEG4-FeAuNPs into the cytosol occurred predominantly in MDA-MB-231 tumor cells but not into PBL. Figure 3B is a 3D model of internalization of TPP-PEG4-FeAuNPs in MDA-MB-231 and PBL cells. The internalization of NPs was also tested by flow cytometry. As shown in Figure 3C, 11.9% of the MDA-MB-231 cells were positively with FITC-labeled TPP-PEG4-FeAuNPs, whereas only 1.47% of the cells were stained after an incubation with FITC-labeled NGL-PEG4-FeAuNPs. In PBL cells, both TPP-PEG4-FeAuNPs and NGL-PEG4-FeAuNPs showed a very low positivity. In Figure 3, we demonstrate that TPP peptide conjugated nanoparticles are more effectively internalized into mHsp70 positive MDA-MB-231 cells.
2) The results are sound but is not clear how is the behavior of the studied systems when using normal cells, epithelial cells could be OK, how is the selectivity, how is the toxicity in normal conditions? Maybe I am wrong but I could not see that in the manuscript.
Answer: Thank you for your suggestion, we chose human peripheral blood lymphocytes (PBL) to test the toxicity. First, we checked the Hsp70 membrane status of PBL cells which showed a mHsp70-positive phenotype on 3.25 ± 0.49% (Figure 1C). In 4T1 (Figure 1A) and MDA-MB-231 (Figure 1B) cells mHsp70-positive phenotype is found on 63.1 ± 5.68% and 61.94 ± 7.88% of the cells respectively. CCK-8 assay was used to determine the toxicity of NGL-PEG4-FeAuNPs and TPP-PEG4-FeAuNPs to PBL. As shown in Figure S2C, a concentration of nanoparticles up to 2.5µg/ml showed no toxicity in PBL. To demonstrate that the internalization of nanoparticles is related to the mHsp70 expression, MDA-MB-231 and PBL were Immunophenotyped for mHsp70. In Figure 3, we show that TPP peptide conjugated nanoparticles are more effectively internalized into mHsp70 positive MDA-MB-231 cells but not in mHsp70 negative PBL.
The authors want to thank you for constructive comments.
Best wishes,
Wu
